# Piezo1 mechanosensing regulates integrin-dependent chemotactic migration in human T cells

**Chinky Shiu Chen Liu[1], Tithi Mandal[2], Parijat Biswas[3], Md Asmaul Hoque[1,4], Purbita Bandopadhyay[1,4], Bishnu Prasad Sinha[1,4], Jafar Sarif[1,4], Ranit D'Rozario[1,4], Deepak Kumar Sinha[3], Bidisha Sinha[2], Dipyaman Ganguly[1]\***

[1]IICB-Translational Research Unit of Excellence, CSIR-Indian Institute of Chemical Biology, Kolkata, India; [2]Department of Biological Sciences, Indian Institute of Science Education and Research, Kolkata, India; [3]Department of Biological Sciences, Indian Association for Cultivation of Science, Kolkata, India; [4]Academy of Scientific and Innovative Research, Ghaziabad, India

**\*For correspondence:**
dipyaman.iicb@gmail.com

**Competing interest:** The authors declare that no competing interests exist.

**Abstract** T cells are crucial for efficient antigen-specific immune responses and thus their migration within the body, to inflamed tissues from circulating blood or to secondary lymphoid organs, plays a very critical role. T cell extravasation in inflamed tissues depends on chemotactic cues and interaction between endothelial adhesion molecules and cellular integrins. A migrating T cell is expected to sense diverse external and membrane-intrinsic mechano-physical cues, but molecular mechanisms of such mechanosensing in cell migration are not established. We explored if the professional mechanosensor Piezo1 plays any role during integrin-dependent chemotaxis of human T cells. We found that deficiency of Piezo1 in human T cells interfered with integrin-dependent cellular motility on ICAM-1-coated surface. Piezo1 recruitment at the leading edge of moving T cells is dependent on and follows focal adhesion formation at the leading edge and local increase in membrane tension upon chemokine receptor activation. Piezo1 recruitment and activation, followed by calcium influx and calpain activation, in turn, are crucial for the integrin LFA1 (CD11a/CD18) recruitment at the leading edge of the chemotactic human T cells. Thus, we find that Piezo1 activation in response to local mechanical cues constitutes a membrane-intrinsic component of the 'outside-in' signaling in human T cells, migrating in response to chemokines, that mediates integrin recruitment to the leading edge.

## eLife assessment

This study provides **useful** insights into the subcellular localization, interaction with integrins, and functional importance of the cell surface receptor Piezo1 in migrating human T-cells. Whether Piezo1 is critically sensing mechano-physical cues during T-cell migration is however not well supported by direct experimental evidence. The data collected is **solid** otherwise.

## Introduction

Efficient T cell migration from blood circulation to secondary lymphoid organs and inflamed tissue is paramount for optimal tissue antigen sampling and effector functions during adaptive immune response (*Shechter et al., 2013*; *Krummel et al., 2016*; *Woodland and Kohlmeier, 2009*; *Masopust and Schenkel, 2013*). The process is dependent on chemotactic cues, the interaction between endothelial adhesion molecules and T cell integrins, and an intricate actomyosin dynamics (*Fowell*

*and Kim, 2021*). Generation of mechanical force is critical to propel migrating cells (*Dupré et al., 2015*; *Nordenfelt et al., 2017*; *Moreau et al., 2018*; *Shannon et al., 2019*). Mechanosensing of the substrate and increase in cellular membrane tension is also intuitively apparent in a migrating T cell, although mechanistic details are not explored to a great extent.

Piezo1 and Piezo2 ion channels are evolutionarily conserved professional mechanosensors that can sense an increase in membrane tension mostly independent of other protein-protein interactions (*Coste et al., 2010*; *Coste et al., 2012*; *Syeda et al., 2016*). Mechanosensing by Piezo1 has been demonstrated to play a critical role in diverse pathophysiologic contexts given its wide tissue expression including the immunocellular compartment (*Murthy et al., 2017*; *Liu et al., 2018*; *Atcha et al., 2021*; *Geng et al., 2021*; *Jairaman et al., 2021*; *Jäntti et al., 2022*; *Liu and Ganguly, 2019*; *Ma et al., 2021*; *He et al., 2022*). We previously identified the crucial role of this specialized mechano-transducer in optimal TCR triggering (*Liu et al., 2018*; *Liu and Ganguly, 2019*). The immunocellular expression of Piezo1 thus makes it a strong candidate for mechanosensing at the plasma membrane to regulate the motility of immune cells. Piezo1 shows differential involvement in cellular migration depending on context (*Huang et al., 2019*; *Chubinskiy-Nadezhdin et al., 2019*; *Mousawi et al., 2020*; *Wang et al., 2021*; *Gao et al., 2021*; *Holt et al., 2021*; *Velasco-Estevez et al., 2022*). Previous studies point to the possibility of a role of Piezo1 in confinement sensing and integrin activation in moving cells, while it deters integrin-independent cell motility, usually referred to as amoeboid movement (*Hung et al., 2016*; *McHugh et al., 2010*).

Here, we aimed at exploring if Piezo1 mechanosensing plays any role during integrin-dependent chemotactic migration of human T cell, a patho-physiologically critical event in an ongoing immune response as well as in the steady-state. We found that deficiency of Piezo1 expression in human CD4[+] T cells as well as Jurkat T cells interfered with efficient integrin-dependent cellular motility in response to a chemotactic cue. We also found that Piezo1 activation constitutes a hitherto unknown membrane-intrinsic component of the 'outside-in' signaling in human T cells in response to chemokine receptor activation, that follows activation of focal adhesion kinase leading to local increase in membrane tension and mediates local recruitment of integrin molecules at the leading edge of the moving human T cells.

## Results

### Piezo1 deficiency abrogates integrin-dependent chemotactic motility in human T cells

We first assessed the role of Piezo1 on integrin-dependent, non-directional motility of primary human CD4[+] T cells. We used GFP plasmid and Piezo1 siRNA co-transfected cells to facilitate the distinction between potentially transfected and untransfected cells during cell tracking. We compared the motility of GFP[+] T cells (presuming they had a good transfection efficiency and thus will also have the siRNA) with the GFP[-] T cells in the same culture. This was confirmed by flow cytometric assessment of Piezo1 protein content on the T cell surface as well as the whole T cells (*Figure 1—figure supplement 1B and C*). Piezo1 deficiency in T cells (potentially GFP[+]) significantly hindered their migration in response to CCL19 in terms of its mean-squared displacement (MSD, *Figure 1A–C*). Representative cell tracks of Piezo1 deficient (GFP[+]) and control (GFP[-]) are shown in *Figure 1B*, *Figure 1—figure supplement 1D* and *Figure 1—animation 1*. These results were further confirmed in a separate experiment where Piezo1 knockdown CD4[+] T cells showed a significant reduction in motility compared to control siRNA-transfected cells (*Figure 1—figure supplement 1E*).

ICAM1-coated transwell chemotaxis assay of human CD4[+] T lymphocytes and Jurkat T cells also showed a significant reduction in migration upon siRNA-mediated Piezo1 knockdown, in response to CCL19 and SDF1α gradients, respectively (*Figure 1D and E*). Piezo1 inhibition with GsMTx4 also led to significant impairment of transwell chemotaxis of CD4[+] T cells (*Figure 1—figure supplement 1F*).

### Redistribution of Piezo1 toward the leading edge of migrating T cells in response to chemotactic cue

Cell polarity is a crucial aspect of directional cell motility in response to chemokines. CD4[+] T cells seeded on ICAM-1 bed showed striking polarity in Piezo1 distribution, upon stimulation with CCL19 for 15 min (*Figure 1F and G*; *Figure 1—animation 2*). Piezo1 m-Cherry expressing Jurkat T cells also

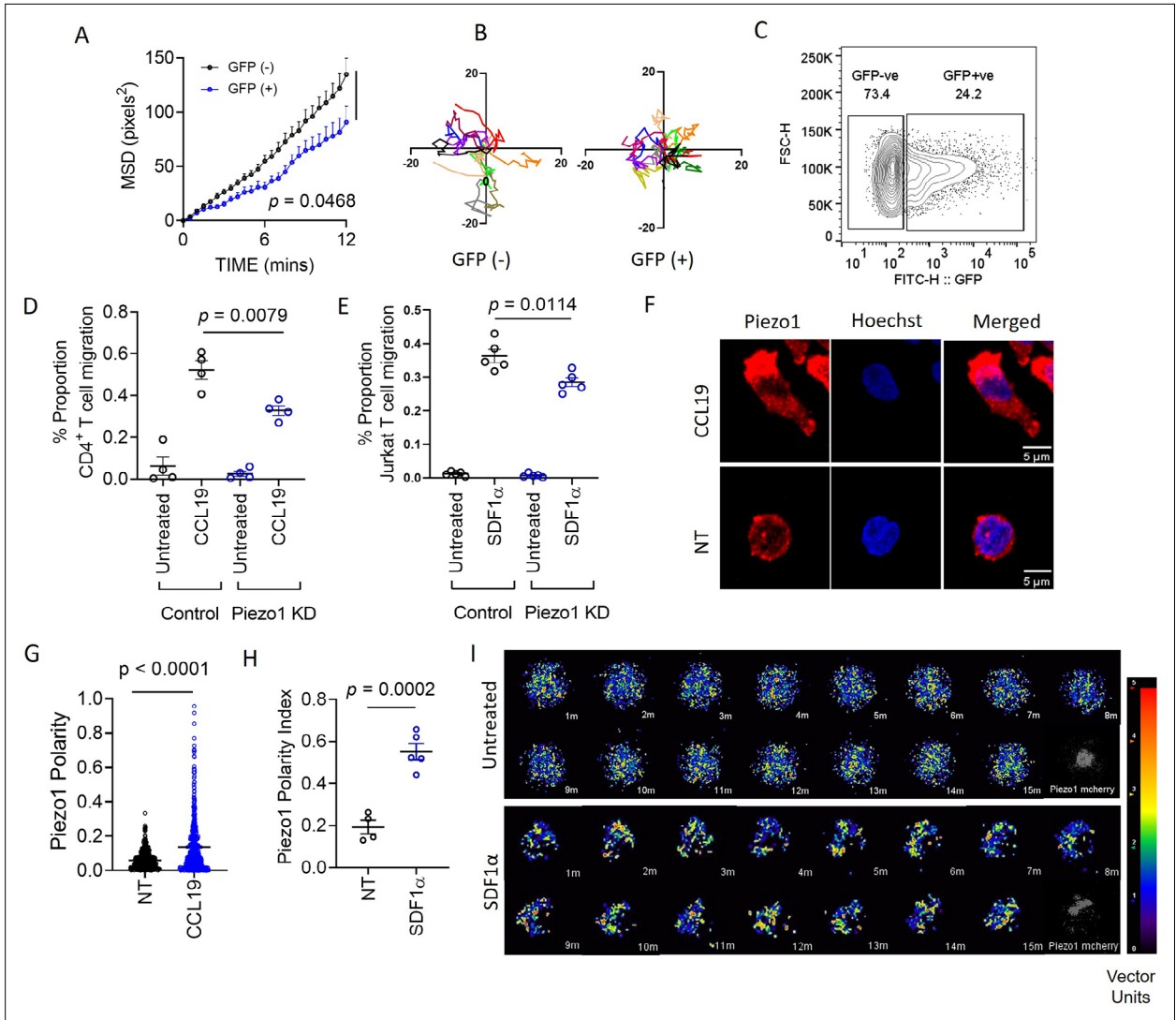

**Figure 1.** Piezo1 deficiency abrogates integrin-dependent motility in human T cells which is mediated through redistribution of Piezo1 at the leading edge in response to chemokine stimulation. (**A**) MSD versus time calculated for GFP$^+$ (potential Piezo1-knockdown cells) and GFP$^-$ (potential control cells) in GFP plasmid and Piezo1 siRNA co-transfected human CD4$^+$ T lymphocytes, that were allowed to migrate in the presence of recombinant CCL19 on ICAM1-coated dishes. (**B**) Representative tracks of GFP$^-$ and GFP$^+$ CD4$^+$ T lymphocytes. (**C**) Comparisons of % GFP$^+$ cells after 72 hr of nucleofection. (**D & E**) 3-D transwell migration assay of siRNA-transfected primary human CD4$^+$ T lymphocytes (**D**) and Jurkat T cells (**E**), respectively. (**F**) Representative confocal images of Piezo1 distribution in fixed untreated and CCL19-treated CD4$^+$ T lymphocytes. 63 X oil magnification. (**G**) Comparison between Piezo1 polarity index calculated for fixed, stained, human CD4$^+$ T lymphocytes with or without 0.5 μg/ml CCL19 treatment for 15 min. n>510 random cells, each. (**H**) Piezo1 polarity index calculated for Jurkat cells, expressing mCherry-tagged Piezo1 during live-cell tracking in the presence of recombinant SDF1α. Each dot represents the polarity index of each cell, averaged over all the time-frames. (**I**) Representative time kinetics of particle image velocimetry (PIV) analysis of Piezo1-mcherry transfected Jurkat cells, allowed to move on ICAM-coated dishes in the presence of recombinant SDF1α. Top panel: No chemokine. Bottom Panel: SDF1α. All data is representative of at least three independent experiments. Student's t-test was used to calculate significance and data is represented as mean ± S.E.M.

The online version of this article includes the following video and figure supplement(s) for figure 1:

**Figure supplement 1.** Piezo1 facilitates integrin-dependent human CD4$^+$ T lymphocyte migration by localizing to the leading edge of cells in response to chemokine stimulation.

**Figure 1—animation 1.** Representative animation of cell trace-stained CD4$^+$ T lymphocytes transfected with GFP construct and Piezo1 siRNA, in the presence of recombinant human CCL19 on ICAM1-coated plates, for a total duration 15 min at 30 s per frame.

**Figure 1—animation 2.** Representative 3D projection of Z-stacks of CD4$^+$ T lymphocyte treated with recombinant human CCL19 for 15 min, immunostained with Piezo1 antibody and Hoechst.

**Figure 1—animation 3.** Representative immunofluorescence time-lapse animation of Piezo1-mCherry expressing Jurkat cell moving on ICAM1-coated dish in the presence of recombinant human SDF1α.

showed striking leading edge polarity in the presence of SDF1α under ICAM1 adhesion (*Figure 1*). In *Figure 1I*, particle image velocimetry (PIV) analysis of a single representative Jurkat cell has been shown to display the dynamically polarized Piezo1 fluorescence upon SDF1α treatment with time, as opposed to rather uniformly distributed Piezo1 signals in a representative untreated cell. The continuous redistribution of Piezo1 within the motile T cells towards the leading edge can also be appreciated in the representative moving image provided (*Figure 1—animation 3*). Redistribution of Piezo1 to the leading edge of chemokine-stimulated CD4⁺ T cells was confirmed from its anti-polar localization to CD44, which preferentially accumulates at the uropod of moving T cells (*Figure 1—figure supplement 1G and H*).

## Piezo1 redistribution in migrating T cells follows increased membrane tension in the leading edge

Piezo1 is a professional mechanosensor ion channel, which senses an increase in plasma membrane tension and responds by driving calcium influx into the cells (*Liu et al., 2018*; *Liu and Ganguly, 2019*; *Xiao, 2020*). As T cells migrate on substrate by extending membrane processes, it is intuitive to hypothesize that a component of mechanosensing is incorporated in the mechanism of cellular movement. The data above, on the critical role of Piezo1 in migrating human T cells, also led us to hypothesize that Piezo1 mechanosensors may play a role in sensing mechanical cues in a moving T cell. So, first we aimed to confirm that there is a polarized increase in tension in the leading edge plasma membrane of human T cells, moving in response to a chemotactic cue.

Cellular membrane tension can be measured by various techniques. However, most of the usual techniques, viz. atomic force microscopy or tension measurement using optical tweezers, are invasive and interfere with the membrane dynamics, as well as are difficult to employ on moving cells. We utilized the non-invasive technique of interference reflection microscopy or IRM for studying the potential redistribution of membrane tension following chemokine stimulation (*Barr and Bunnell, 2009*; *Chakraborty et al., 2022*). In the IRM technique, steady-state local fluctuations in the plasma membrane are imaged in cells adhered to a glass coverslip. The interference patterns generated due to continuously changing local distance between the coverslip and the plasma membrane are computed to derive the magnitude of local tension in the membrane – lower temporal fluctuations represent an increase in membrane tension (*Figure 2A*). After baseline imaging, Piezo1-GFP expressing Jurkat cells were subjected to sequential fluorescence and IRM imaging post SDF1α addition which was followed over time. Chemokine administration to Jurkat T cells led to a decrease in temporal fluctuations and an increase in membrane tension when computed for the whole cellular contour (*Figure 2B*). The increase in membrane tension was registered within a few minutes after chemokine addition and gradually declined over time (*Figure 2C*). Distribution of tension and fluctuation-amplitude in cells was also affected by the action of chemokine (*Figure 2—figure supplement 1A*) which was accompanied by an increase in the membrane's degree of confinement and the effective viscosity experienced (*Figure 2—figure supplement 1C and D*). Moreover, the tension maps showed that the lamellipodial structures towards the leading edges of the cells had higher tension magnitudes (*Figure 2D*, *Figure 2—figure supplement 1B*). These high-tension edges are usually further emphasized at later time-points.

Finally, we explored if local membrane tension increase was linked to Piezo1 redistribution in the chemokine-experienced T cells. Of note here, during acquisition, there is typically a time delay of 1 min between measurement of tension and Piezo1-GFP fluorescence. As expected, average tension peaked within the first few minutes of chemokine addition, followed by a partial decline (*Figure 2E*) as seen for the whole cell. Pixel count of Piezo1-GFP intensity showed a similar pattern, albeit, at a delayed time, following the tension kinetics. In order to account for the delayed kinetics of tension magnitude and Piezo1-GFP pixel count, we performed a correlation between normalized Piezo1 pixel intensity at a particular time-frame, with normalized tension magnitude at the preceding time frame. We observed a significant positive correlation between the two cell-averaged parameters (*Figure 2F and G*). This, however, does not capture the local spatial correlation between Piezo-1 intensity and tension. Thus, IRM-coupled fluorescence microscopy revealed an increase in local membrane tension in the leading edge of the moving T cells in response to chemokine receptor activation that was associated with Piezo1 redistribution to the leading edges of the cells.

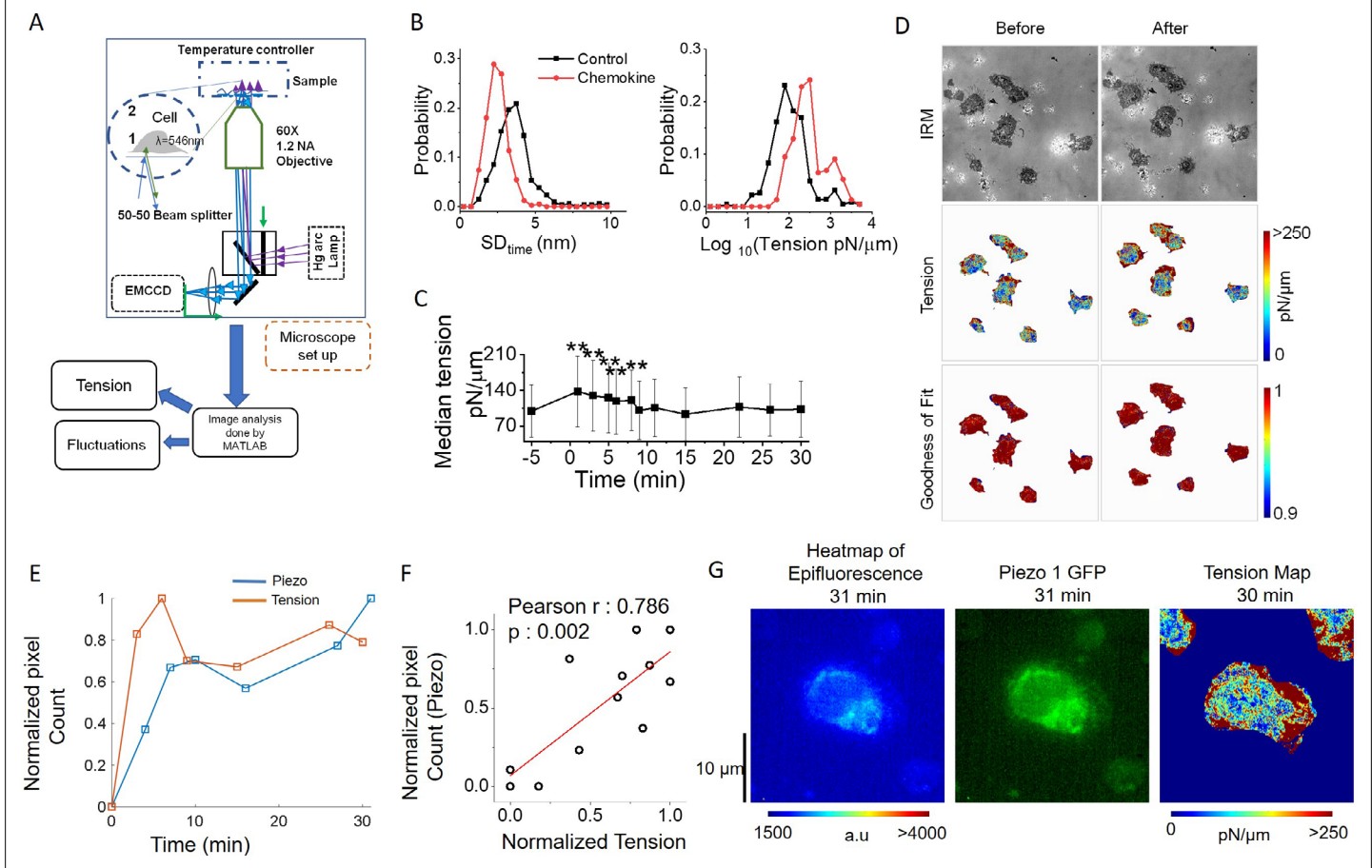

**Figure 2.** Piezo1 redistribution in migrating T cells follows increased membrane tension in the leading edge. (**A**) Schematic illustration of interference reflection microscopy imaging setup. (**B**) Probability distribution of the amplitude of temporal fluctuations (SD$_{time}$, left) and tension (right) before and after the addition of chemokines. (**C**) Temporal trajectory of tension of cells before and after addition of chemokine with ** indicating a significant difference from control. (**D**) Representative IRM images of Jurkat cells (top), corresponding tension maps (middle), and R$^2$ maps (bottom), before and after 3 min of chemokine treatment. (**E**) Correlation between normalized Piezo intensity, tension magnitudes with time. (**F**) Scatter plot to show the correlation between the normalized pixel count of Piezo1 and normalized tension. Piezo1 intensity at specific time-points were correlated with tension magnitudes at the preceding timepoints. Normalization has been done such that the maximum value is set to 1 and others are accordingly scaled. (**G**) Colour-coded Piezo1-GFP intensity map (middle), epifluorescence image (left), and tension map (right) of representative Piezo1-GFP expressing Jurkat cell, after 30 min of 0.1 µg/ml of SDF1α treatment.

The online version of this article includes the following figure supplement(s) for figure 2:

**Figure supplement 1.** Chemokine stimulation increases leading edge membrane tension in human CD4[+] T lymphocytes.

## Focal adhesion following chemokine receptor activation does not depend on Piezo1

Focal adhesion formation is a critical step in cellular migration and activated phosphorylated focal adhesion kinases link extracellular physical cues to the cytoskeletal actomyosin scaffold (*Mitra et al., 2005*; *Katoh, 2020*). Chemokine signaling induces phosphorylation of focal adhesion kinase (FAK) and membrane recruitment of the FAK complex forming focal adhesions with the extracellular substrate. FAK recruits adapter proteins like paxillin to anchor with the local sub-membrane cytoskeleton made by de novo F-actin polymerization. Membrane-recruited FAK complexes anchored to the cytoskeleton increases local membrane tension as well as recruit integrins to the membrane (*Yamashiro and Watanabe, 2014*; *Dupré et al., 2015*; *Martino et al., 2018*). This so-called 'outside-in' signaling, whereby membrane recruitment of integrins (e.g. LFA1 in T cells) is initiated, is followed by integrin signaling ('inside-out' signaling) driven primarily by phosphoinositide 3-kinase (PI3K)-driven phosphorylation of Akt and downstream activation of Rho GTPases which in turn drive extensive F-actin polymerization (*Sánchez-Martín et al., 2004*; *Roy et al., 2020*). Retrograde flow of the F-actin to the rear end of the

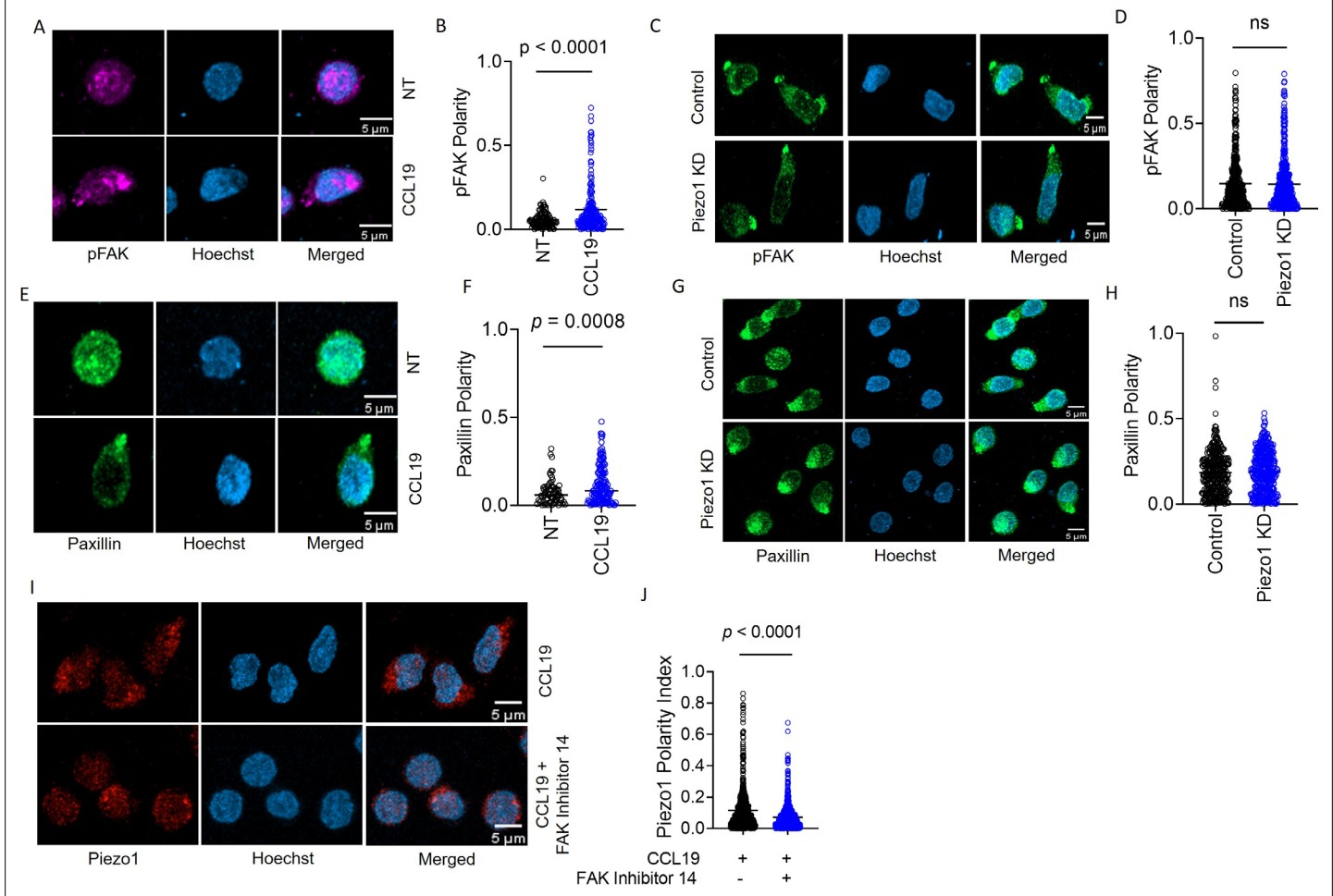

**Figure 3.** Focal adhesion following chemokine receptor activation does not depend on Piezo1. (**A**) Representative confocal images of human CD4+ T lymphocytes, fixed and stained for phospho-FAK (pFAK) under untreated and CCL19-treated conditions. (**B**) Increased polarity of pFAK upon chemokine stimulation of CD4+ T lymphocytes as compared to unstimulated controls. n>120 cells, each. (**C**) Representative confocal images of pFAK distribution in control and Piezo1 siRNA-transfected CD4+ T lymphocytes stimulated with chemokine. (**D**) Comparison of pFAK polarity in CCL19-stimulated control and Piezo1-knockdown CD4+ T lymphocytes. n>450 random cells, each. (**E**) Representative confocal images of paxillin distribution in untreated versus CCL19-stimulated CD4+ T lymphocytes. (**F**) Increased paxillin polarity in response to CCL19 stimulation as compared to untreated CD4+ T lymphocytes. Untreated n>70, CCL19-treated n>150. (**G**) Representative confocal images of immunostained paxillin in chemokine-stimulated, control and Piezo1-knockdown CD4+ T lymphocytes. (**H**). Comparison of paxillin polarity in CCL19-stimulated control and Piezo1-knockdown CD4+ T lymphocytes. n>550 random cells, each. (**I**) Representative confocal images of stained Piezo1 in CD4+ T cells stimulated with CCL19 in the presence or absence of FAK inhibitor 14. (**J**) Effect of focal adhesion kinase (FAK) inhibition on Piezo1 polarity in CD4+ T lymphocytes stimulated with recombinant CCL19 versus untreated cells. n>600 random cells, each. All data represented is from at least three independent experiments. Student's t-test was used to calculate significance and data is represented as mean ± S.E.M.

The online version of this article includes the following video and figure supplement(s) for figure 3:

**Figure supplement 1.** Piezo1 acts downstream of focal adhesion formation and localizes to sites of focal adhesions formed in response to chemokine stimulation.

**Figure 3—animation 1.** Representative 3D rendering of confocal Z-stacks of CD4+ T lymphocyte treated with recombinant human CCL19 for 10 min, immunostained with phosphorylated focal adhesion kinase (pFAK) antibody and Hoechst.

moving cells, through myosin-based contractions, generates the propelling force for cells to move forward following the chemokine gradient (*Dupré et al., 2015*; *Nordenfelt et al., 2017*).

Finding a critical role of Piezo1 in human T cell migration and a membrane tension-dependent redistribution of Piezo1 to the leading edge of the moving cells led us to explore if FAK recruitment to the leading edge in response to chemokine stimulation was dependent on Piezo1. While phosphorylated FAK (pFAK) showed striking polarity upon CCL19 stimulation on ICAM-1 bed (*Figure 3A and B*,

*Figure 3—animation 1*), Piezo1 knockdown did not have any effect on pFAK polarity under the same conditions (*Figure 3*). We observed similar results upon examining the distribution of paxillin which is another critical component of focal adhesions. Paxillin showed increased polarity upon CCL19 stimulation (*Figure 3E and F*) and its polarity remained unaffected upon Piezo1 knockdown (*Figure 3G and H*). We also did not observe any changes in pFAK and paxillin signal intensity upon Piezo1 knockdown in CCL19-stimulated cells (*Figure 3—figure supplement 1A and B*, respectively). However, pFAK showed striking co-localization with Piezo1 upon CCL19 stimulation (*Figure 3—figure supplement 1C*). Thus, focal adhesion formation following chemokine receptor activation was not dependent on Piezo1. FAK activation recruits adapters like paxillin and vinculin connecting the cortical cytoskeleton to the membrane and the extracellular matrix, thereby increasing local membrane tension (*Mitra et al., 2005*). Inhibition of FAK activation completely abrogated Piezo1 redistribution to the leading edge, hence the polarity (*Figure 3I and J*). This indicated that mechanistically, the FAK activation event precedes Piezo1 redistribution to the leading edge of the migrating human CD4$^+$ T cells.

## Membrane recruitment of LFA1 on chemokine receptor activation disrupts with Piezo1 deficiency

Integrin recruitment to the leading edge occurs downstream of FAK activation and local F-actin polymerization. As expected, CCL19 stimulation led to LFA1 recruitment to the leading edge upon CCL19 stimulation on the ICAM1 bed (*Figure 4A and B*; *Figure 4—animation 1*). Moreover, Piezo1 and LFA1 (CD11a) exhibited significant colocalization upon chemokine stimulation (*Figure 4*). Piezo1 downregulation abrogated LFA-1 polarity (*Figure 4D and E*; *Figure 4—animation 2 and 3*). Thus, the role of Piezo1 in chemotactic migration of T cells was downstream of FAK assembly in response to chemokine receptor signaling, but preceded integrins recruitment. These data together point to a mechanism whereby local increase in membrane tension leading from FAK activation and assembly drives Piezo1 activation, which in turn is linked to LFA1 recruitment to the focal adhesions.

To further confirm this critical role of Piezo1 on LFA1 recruitment, we looked at the major signaling event downstream of LFA1 recruitment to the leading edge, i.e., phosphorylation of Akt downstream of PI3K activation, which in turn is triggered upon integrin activation. On treatment with CCL19, human CD4$^+$ T cells moving on ICAM-1-coated plates did show a prominent leading-edge polarity of non-phosphorylated and phosphorylated Akt (pAkt) (*Figure 4F*, control panel). But in Piezo1-deficient cells, the polarity of pAkt is abrogated (*Figure 4G*), not affecting the polarity of non-phosphorylated Akt (*Figure 4H*; *Figure 4—animation 4 and 5*), substantiating the cellular signaling defect due to non-recruitment of LFA1 to the leading edge membrane.

## Piezo1 deficiency disrupts F-actin retrograde flow in T cells despite chemokine receptor activation

Piezo1-mediated sensing of membrane tension drives calcium ion influx, which in turn has been shown to activate calpain and F-actin polymerization (*Liu et al., 2018*; *Liu and Ganguly, 2019*). LFA1 recruitment is driven by local actin scaffold formation following FAK assembly, which as our data indicates is perhaps contributed by consolidation of local actin scaffold downstream of Piezo1 activation. To confirm this, we performed Ca$^{2+}$ imaging in Piezo1-mCherry transfected Jurkat Cells using a total internal reflection fluorescence (TIRF) microscope in response to chemokine. We found significant colocalization of the Ca$^{2+}$ signal with the Piezo1 signal on the chemokine-treated Jurkat cells (*Figure 5A and B*; *Figure 5—figure supplement 1A and B*). Next, to confirm that activation of calpain mediates the Piezo1 function, we observed that prior inhibition with calpain inhibitor PD150606 abrogated CCL19-induced LFA1 recruitment to the leading edge of the T cells (*Figure 5C and D*). As expected, the upstream event of pFAK polarity was not affected by the calpain inhibition (*Figure 5—figure supplement 1C and D*).

While F-actin polymerization downstream of FAK is required for LFA1 recruitment, LFA1 in turn drives a more robust F-actin polymerization at the leading edge, through Akt/Rho GTPase activation. Leading edge F-actin undergoes retrograde flow due to myosin contractions, thereby enabling cells to move forward. Thus, both at the leading edge lamellipodia and the rear edge F-actin redistribution can be documented in a motile cell. We observed focal co-localization of LFA1 with F-actin at the leading edge, along with a denser actin-rich protrusion in the rear edge or the uropod (*Figure 5C*), which was abolished upon calpain inhibition (*Figure 5C and E*). Hence, LFA1 recruitment may depend

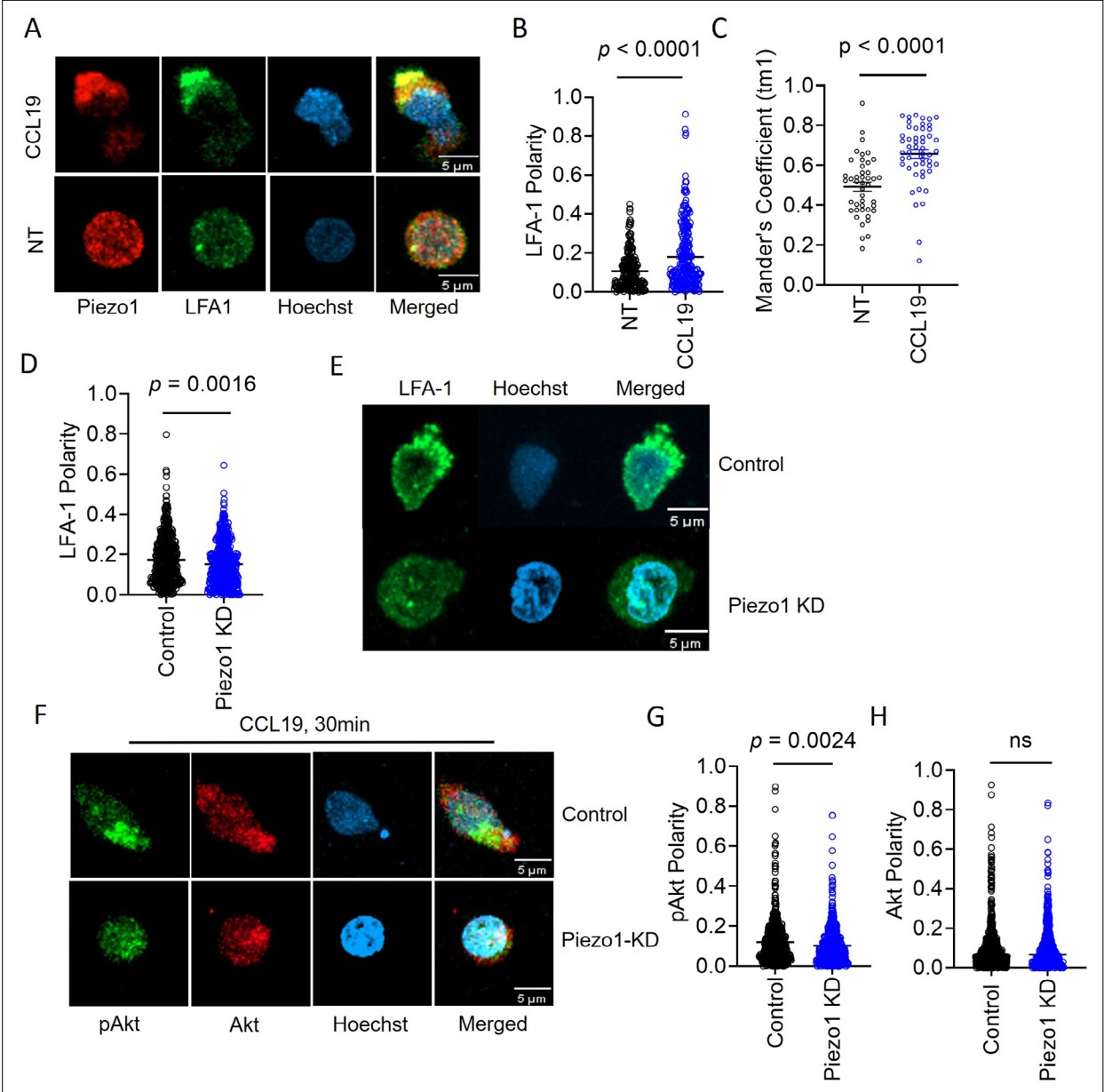

**Figure 4.** Membrane recruitment of LFA1 on chemokine receptor activation disrupts Piezo1 deficiency. (**A**) Representative confocal images of fixed, immunostained Piezo1 and LFA1 in unstimulated and CCL19-stimulated CD4[+] T lymphocytes. (**B**) Increased LFA1 polarity in response to recombinant CCL19 stimulation in CD4[+] T cells. n>295 random cells, each. (**C**) Manders' co-localization analysis of Piezo1 and LFA1 in untreated and CCL19-treated CD4[+] T lymphocytes. n>100 cells, each. (**D**) Comparison of LFA1 polarity in chemokine-treated control and Piezo1 knockdown CD4[+] T lymphocytes. n>490 random cells, each. (**E**) Representative confocal stained images of LFA1 polarity of CCL19-treated control and Piezo1-knockdown CD4[+] T lymphocytes. (**F**) Representative confocal images of phospho-Akt (pAkt) and Akt distribution upon chemokine treatment of control and Piezo1-knockdown cells. (**G** & **H**) Quantitative analyses of pAkt (**G**) and Akt (**H**) polarity in control and Piezo1-knockdown cells after CCL19 treatment. n>580 random cells, each. All data is representative of at least three independent experiments. Student's t-test was used to calculate significance and data is represented as mean ± S.E.M.

The online version of this article includes the following video(s) for figure 4:

**Figure 4—animation 1.** Representative 3D rendering of confocal Z-stacks of CD4[+] T lymphocyte depicting Piezo1 and CD11a/LFA-1 distribution, upon 30 min of recombinant CCL19 treatment.

**Figure 4—animation 2.** Representative 3D rendering of confocal Z-stacks of control siRNA-transfected CD4[+] T lymphocyte depicting CD11a/LFA-1 polarity, upon 30 min of recombinant CCL19 treatment.

**Figure 4—animation 3.** Representative 3D rendering of confocal Z-stacks of Piezo1 siRNA-transfected CD4[+] T lymphocyte depicting CD11a/LFA-1

*Figure 4 continued on next page*

*Figure 4 continued*

polarity, upon 30 min of recombinant CCL19 treatment.

**Figure 4—animation 4.** Representative 3D rendering of confocal Z-stacks of control siRNA-transfected CD4[+] T lymphocyte depicting phosphorylated Akt polarity, upon 30 min of recombinant CCL19 treatment.

**Figure 4—animation 5.** Representative 3D rendering of confocal Z-stacks of Piezo1 siRNA-transfected CD4[+] T lymphocyte depicting phosphorylated Akt polarity, upon 30 min of recombinant CCL19 treatment.

on local F-actin polarization downstream of Piezo1 activation. On the other hand, failure of LFA1 recruitment and activation halts further F-actin polarization feeding the retrograde flow of actin during migration.

Dichotomy of Piezo1 (dense leading edge distribution) and F-actin (more dense at the rear end) distribution was clearly demonstrated in cells stimulated with chemokine (*Figure 6A* and *Figure 6—animation 1*). But, no F-actin polarity was seen on downregulating Piezo1 in the T cells (*Figure 6*; *Figure 6—animation 2* and *Figure 6—animation 3*). Live cell imaging of Jurkat T cells expressing Piezo1-mCherry and actin-GFP on ICAM-1 coated dishes was performed in the presence or absence of SDF1α (*Figure 6—figure supplement 1*). Chemokine addition showed a significant increase in Piezo1 polarity to the front end, as shown in the earlier experiments (*Figure 6E*, *Figure 6—figure supplement 1B*, left) upon chemokine stimulation when compared to untreated cells (*Figure 6D*, *Figure 6—figure supplement 1B*, left). But F-actin showed a dynamic balance at both ends as expected (*Figure 6E*, *Figure 6—figure supplement 1B*, right). Polar plots depicting the angular distribution of Piezo1 mCherry and actin-GFP (over all time-points) in the presence or absence of SDF1α are shown in *Figure 6—figure supplement 1C*. Representative *Figure 6F* and *Figure 6—animation 4* show Piezo1-mCherry distribution predominantly at the leading edge and actin-GFP eventually accumulating at the rear end in uropod-like cellular extensions in a migrating Jurkat T cell.

Thus, our data reveals a membrane-intrinsic event, involving a critical role of Piezo1 mechanosensors, within the outside-in signaling module in response to chemokine receptor activation in human T cells. The mechanistic model derived from our data is depicted in *Figure 7*. The Piezo1 mechanosensing links the focal adhesion assembly to integrins recruitment at the leading edge of the human T cells migrating in response to chemokine activation and thus plays a critical role in the chemotactic migration of human T cells.

## Discussion

The present study explored the role of membrane-intrinsic tension generation and the crucial role of sensing this physical cue in the context of human T cell migration in response to chemokine receptor activation. Deficiency of Piezo1 inhibited integrin-dependent migration, demonstrating the crucial role of these mechanosensors in this mode of T cell migration. As the integrin-dependent chemotactic migration of T cells is critically implicated in most pathophysiologic contexts, in the present study we focused on the potential role of Piezo1 mechanosensors in this specific context.

Notably, a previous study in vivo in mice, using genetic deletion of Piezo1 in T cells, reported no significant effect on tissue recruitment of T cells (*Jairaman et al., 2021*). Although, a functional redundancy in the role of Piezo1-mediated mechanosensing in mouse T cells warrants exclusion, especially in case of gene deletion in developing T cells, given the considerable expression of Piezo2 as well in mouse T cells (*Ganguly, 2021*). On the other hand, the role of Piezo1 activation in fibroblastic reticular cells in Peyer's patches has been shown to regulate lymphocyte migration through tissue conduits (*Chang et al., 2019*). The different tissue distribution of Piezo1 and Piezo2 expression in humans warranted exploring the role of Piezo1 in migrating human T cells, as in humans Piezo2 is hardly expressed in the hematopoietic cells.

Non-invasive interference reflection microscopy allowed us to demonstrate dynamic change in membrane tension at the leading edge of T cells stimulated with chemokine on ICAM1-bed which was closely followed by dynamic recruitment Piezo1 locally. Chemokine receptor activation leads to focal adhesion formation at the leading edge of the migrating T cells (*Mitra et al., 2005*; *Katoh, 2020*). Our experiments revealed that Piezo1 recruitment at the leading edge of the cells followed focal adhesion kinase activation. Focal adhesions do increase local membrane tension as they connect the extracellular environment with the cortical cytoskeleton of the cell (*Mitra et al., 2005*). Piezo1 recruitment

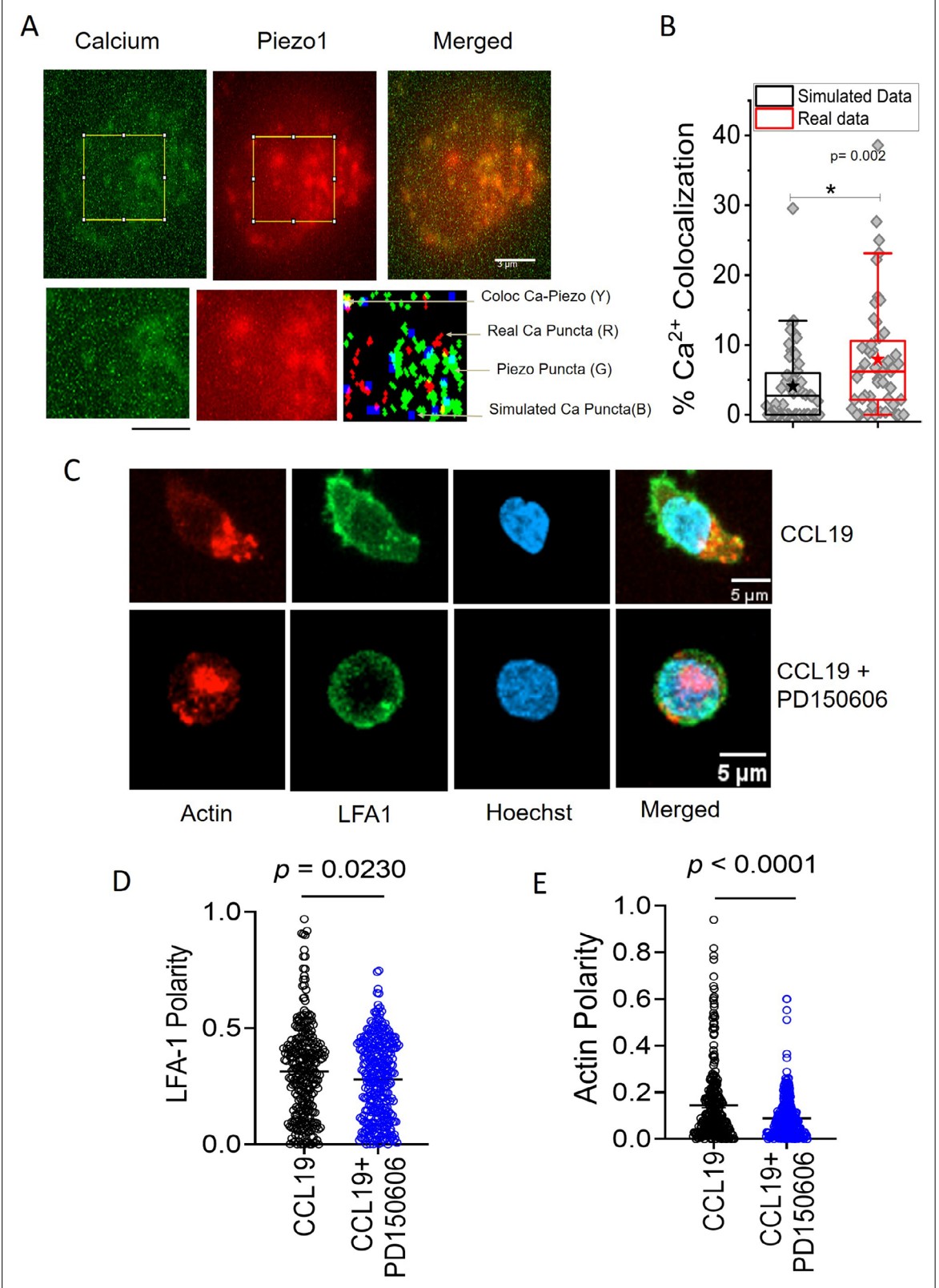

**Figure 5.** Local Ca²⁺ mobilization with Piezo1 redistribution upon chemokine stimulation. (**A**) Representative total internal reflection fluorescence (TIRF) image of a Jurkat Cell stained with Fluo-3, AM and transiently expressing with Piezo1 mCherry. Lower panel: zoomed-in image and overlap of binary images of objects detected from Calcium and Piezo channels as well as simulated randomly placed punctas having similar total (in the zoomed-in section) and average area as that of the Calcium punctas detected in the real images. Overlap RGB image shows the overlap between real

*Figure 5 continued on next page*

*Figure 5 continued*

calcium (red) and piezo puncta (green), overlap puncta (yellow), and simulated calcium puncta (blue). (**B**) Comparison between percentage calcium colocalization of simulated data and real data. * Denote $p < 0.05$ calculated using Mann Whitney U test. n cell = 52 (13 cells x 4 time points). NROIS = 52. (**C**) Representative confocal images of LFA1 and actin distribution in CCL19-stimulated CD4 + T lymphocytes with or without inhibition of calpain. 100 µM of PD15606 was added to the cells 1 hr prior to the addition of chemokine. Quantitative comparisons of (**D**) LFA1 and (**E**) actin polarity upon chemokine stimulation, in the presence or absence of calpain pre-inhibition. n>290 random cells, each. All data is generated from at least three independent experiments. Student's t-test was used to calculate significance and data is represented as mean ± S.E.M.

The online version of this article includes the following figure supplement(s) for figure 5:

**Figure supplement 1.** Active calcium mobilization indicating Piezo1 activity at Piezo1-rich areas of focal adhesions formed independently of Piezo1-mediated actin polymerization.

locally seems to be a response to this membrane tension increase at focal adhesions. Piezo1 recruitment to focal adhesions was also shown to play a role in enhancing glioblastoma aggression by activating integrin-FAK signaling upstream of ECM remodeling and tissue stiffening (*Chen et al., 2018*). Interestingly, a recent study by Yao et. al., showed that Piezo1 is required for the formation of mature focal adhesions. Piezo1 binds to matrix adhesion regions in a force-dependent manner via a linker domain and this binding is abrogated in cancer cells causing transformation to malignant phenotypes (*Yao et al., 2022*).

'Outside-in' signaling downstream of focal adhesion formation recruits integrins to the leading edge of the moving T cells (*Katoh, 2020*; *Yamashiro and Watanabe, 2014*) which was abrogated by Piezo1 downregulation. Piezo1 has been shown to drive $Ca^{2+}$ influx into the T cells leading to calpain activation (*Liu et al., 2018*; *Liu and Ganguly, 2019*). As expected, calpain inhibition also interfered with leading-edge recruitment of the integrin LFA-1. Plausibly calpain activation drives local F-actin polymerization at the leading edge facilitating LFA-1 recruitment. This was also supported by the loss of polarity of Akt phosphorylation with Piezo1 deficiency, which is known to be the major downstream signaling module that succeeds LFA1 recruitment and activation (*Roy et al., 2020*; *Roy et al., 2018*). These dysregulations in turn abrogates the retrograde flow of actin in the moving cell that is essential for forward propulsion of the cell body (*Dupré et al., 2015*; *Nordenfelt et al., 2017*; *Martino et al., 2018*; *Yao et al., 2022*). Of note here, the role of ER-resident Piezo1, in driving R-RAS-driven activation of integrins, which was again dependent on calpain activation, was reported even before the mechanosensing function of Piezo1 was established (*McHugh et al., 2010*).

Thus, the present study reports a hitherto unexplored role of the professional mechanosensor Piezo1 in integrin-dependent chemotactic movement of human T cells. In the cascade of leading edge events in a moving human T cell, following chemokine receptor activation and eventually leading to the retrograde flow of actin, focal adhesion formation precedes Piezo1 activation, while LFA-1 recruitment is dependent on Piezo1 function.

## Materials and methods

### Human CD4[+] T cell isolation and transfection

Details of reagents used in the study are mentioned in the key resources table (Appendix 1).

Human CD4[+] T lymphocytes were isolated from peripheral blood from healthy donors through magnetic immunoselection. Peripheral blood samples from healthy human subjects were collected on obtaining written informed consents. The studies were approved by the Human Ethics Committee of CSIR-Indian Institute of Chemical Biology, India. Cells were transfected with either EGFP control or Piezo1-specific siRNAs using Lonza 4D nucleofector, as per the manufacturer's protocol. Briefly, cells were nucleofected in 100 µl of supplemented P3 primary buffer containing 175 ng of respective siRNAs. Nucleofection was performed using an EO-115 pulse in Lonza 4D nucleofector. Nucleofected cells were incubated at 37 °C. 5% $CO_2$ in RPMI media containing 10% FBS for 3 days before experiments were conducted. Knockdown was checked by qPCR measurement of Piezo1 and Piezo2 transcript levels (*Figure 1-figure supplement 1A*).

### GFP/Piezo1 siRNA co-nucleofection

Approximately $2 \times 10^6$ human CD4 + T lymphocytes were nucleofected with 3 µg of GFP-expressing plasmid along with 175 ng of Piezo1-specific siRNA. Cells were incubated for 3 days before migration

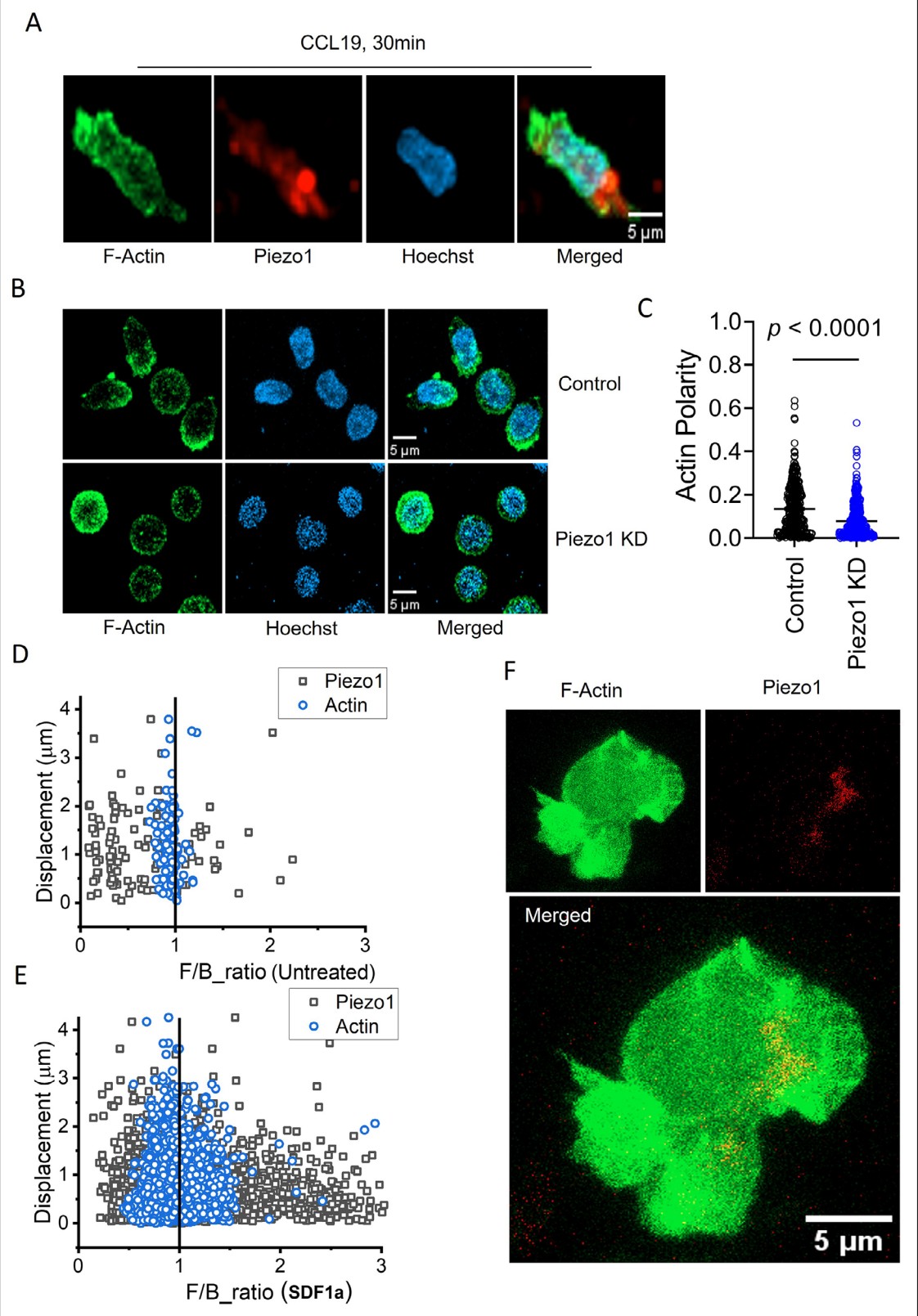

**Figure 6.** Piezo1 deficiency disrupts F-actin retrograde flow in T cells despite chemokine receptor activation. (**A**) Representative fixed confocal images of actin and Piezo1 distribution in CD4+ T lymphocytes after 30 min of 0.5 µg/ml recombinant CCL19 treatment. (**B**) Representative fixed confocal images of actin distribution in chemokine-treated, control and Piezo1-knockdown CD4 + T lymphocytes. (**C**) Quantitative comparison of actin polarity in control and Piezo1-knockdown CD4+ T lymphocytes cells after 30 min of chemokine treatment. n>300 random cells, each. Front-back (F/B) of Piezo1

*Figure 6 continued on next page*

*Figure 6 continued*

and actin-GFP in untreated (**D**) and SDF1α treated (**E**) Jurkat cells co-expressing actin-GFP and Piezo1-mCherry. (**F**) A snapshot of time-lapse imaging of Piezo1-mCherry and actin-GFP expressing Jurkat cell treated with 0.1 μg/ml of SDF1α. Image is the maximum Z-projection of the cell at 63 X/1.40 magnification. All data is generated from at least three independent experiments.

The online version of this article includes the following video and figure supplement(s) for figure 6:

**Figure supplement 1.** Leading edge polarity of Piezo1-mCherry and dynamic distribution of actin-GFP in SDF1α-treated Jurkat cells.

**Figure 6—animation 1.** Representative 3D rendering of confocal Z-stacks of CD4⁺ T lymphocyte depicting Piezo1 and actin distribution, upon 30 min of recombinant CCL19 treatment.

**Figure 6—animation 2.** Representative 3D rendering of confocal Z-stacks of control siRNA-transfected CD4⁺ T lymphocyte depicting actin polarity, upon 30 min of recombinant CCL19 treatment.

**Figure 6—animation 3.** Representative 3D rendering of confocal Z-stacks of Piezo1 siRNA-transfected CD4⁺ T lymphocyte depicting loss of actin polarity, after 30 min of recombinant CCL19 treatment.

**Figure 6—animation 4.** Representative time-lapse animation of Piezo1-mCherry/actin-GFP expressing Jurkat cell, moving on ICAM1-coated dish in the presence of recombinant human SDF1α.

studies. EGFP esiRNA was used as control siRNA in experiments involving a comparison between control and Piezo1 siRNA-KD cells.

## Analysis of Piezo1 knockdown

GFP and Piezo1 siRNA co-transfected CD4⁺ T cells were stained for Piezo1 to measure downregulation. Both surface and intracellular (0.2% triton X-100 permeabilization) Piezo1 were stained. Briefly, cells were collected on the day and fixed in 0.5% PFA for 5 min on ice. Cells were stained with 1.67 μg/ml of rabbit anti-Piezo1 for 45 min at 4 °C. After washing, 1.5 μg/ml of anti-rabbit alexa 568-IgG was added and incubated for 30 min at 4 °C. Cells were acquired on BD LSR Fortessa III after washing. Piezo1 mean fluorescence intensity (MFI) was measured both for GFP⁺ and GFP⁻ cells. Paired t-test was used to calculate statistical significance.

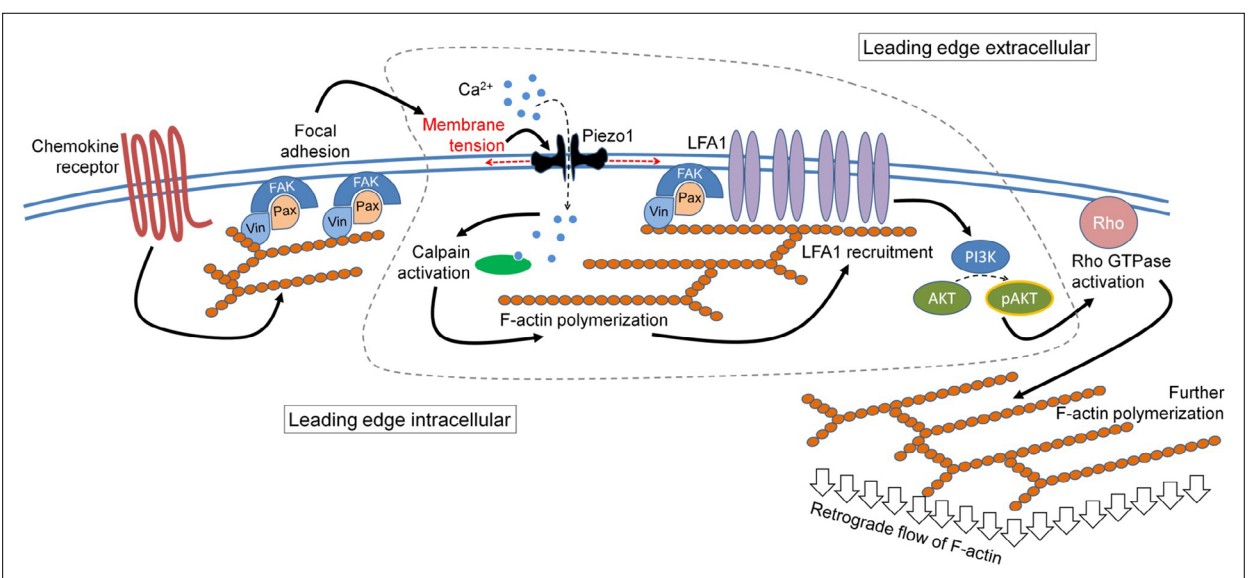

**Figure 7.** The mechanistic model depicting the involvement of Piezo1 mechanosensing in leading-edge events in a migrating T cell. Proposed model suggests chemokine receptor activation in human T cells leads to focal adhesion kinase activation and focal adhesion formation. Focal adhesions lead to localized increase in membrane tension at the leading-edge plasma membrane which leads to Piezo1 recruitment and activation. Piezo1 activation leads to calpain activation which potentially drives further cytoskeletal consolidation to recruit integrin LFA1. LFA1 recruitment and activation lead to phosphorylation of AKT and downstream signaling eventually driving the retrograde actin flow in migrating human T cells.

## Quantitative PCR analysis of Piezo1 knockdown

Transfected CD4+ T cells were analyzed for Piezo1 knockdown using primers against human Piezo1 and Piezo2. Primers specific to housekeeping gene, 18 S was used to calculate the relative expression of Piezo1 and Piezo2. Paired t-test was used to calculate statistical significance.

## 2-dimensional cell tracking

12 mm confocal dishes were coated with 4 µg/ml of recombinant human ICAM-1 overnight at 4 °C. Wells were washed with 1 X PBS. Tracking of GFP/Piezo1 siRNA-transfected cells were performed after cell trace violet-labeling in the presence of recombinant CCL19 on ICAM1-coated dishes. Stained cells were added to the wells and allowed to settle for 1 hr at 37 °C. 2D tracking was performed at 37 °C after the addition of 0.5 µg/ml of recombinant human CCL19 and incubated for 10 min before acquisition. Images were acquired at regular intervals at room temperature using a Zeiss LSM setup at 20 X magnification for a total duration of 10–15 min at 30 s interval. CD4+ T lymphocytes were RPMI without phenol red supplemented with 10% FBS and 25 mM HEPES during the duration of imaging.

## 2D cell tracking analysis

For tracking analyses of GFP and Piezo1 siRNA co-transfected cells, all cells above a specific threshold area were used to eliminate potential debris and dead cells due to GFP plasmid transfection. Time-based image stacks obtained were analyzed using the Particle Tracker 2D/3D plugin of the FIJI Mosaic Suite. Briefly, images were corrected for any shift in alignment during acquisition, using the StackReg plugin. Corrected images were thresholded to create a 0–255 binary mask which were then used for tracking using the Particle Tracker 2D/3D plugin. Tracking provided frame-wise X-Y co-ordinates of cells which were subsequently used to derive the following parameter to measure motility:

a. Mean-squared displacement

$$MSD(n) = \sum_{i=0}^{i=n} (x_n - x_0)^2 + (y_n - y_0)^2$$

where, i=0 is the first frame, i=n is the nth frame.

## Tranwell chemotaxis assay

5 µm transwell inserts were coated with 4 µg/ml of recombinant human ICAM-1. Equal number of primary human CD4+ T cells and Jurkat cells nucleofected with control or Piezo1-specific siRNA were added to the ICAM-1-coated inserts. Cells were allowed to migrate in response to recombinant human CCL19 (0.5 µg/ml) or SDF1α (0.1 µg/ml) in the lower chamber wells for 5 hr and 1.5 hr, respectively, at 37 °C, 5% $CO_2$. Migration was halted by placing the setup on ice. Percent migration was calculated by counting the number of cells in the lower chamber and dividing it by the total number of cells that was seeded.

## Overexpression of mCherry/GFP-tagged Piezo1 and actin-GFP in Jurkat cells

GFP and mCherry-tagged Piezo1 constructs were kindly provided by Dr. Charles Cox at Victor Chang Cardiac Research Laboratory, Darlinghurst, Australia. Approximately 2 million Jurkat cells were nucleofected with 2–3 µg of fluorescent Piezo1 construct in 100 µl of supplemented SE cell line buffer. Cells were pulsed using the CL-120 program in Lonza 4D nucleofector. Cells were cultured in antibiotics-deficient RPMI supplemented with 20% FBS for 36 hr. Overexpression was confirmed using flow cytometry.

## Interference reflection microscopy and analysis

Jurkat cells were nucleofected with 2–3 µg/ml of Piezo1-GFP construct. Overexpression was confirmed using flow cytometry. Transfected cells were seeded on recombinant human fibronectin glass-based dishes. Cells were allowed to completely adhere for 1 hr to allow IRM imaging.

An inverted microscope (Nikon, Japan) with 60 × 1.2 NA water-immersion objective, Hg arc lamp, and necessary interference filter (546±12 nm) were used for the interference reflection microscopy (*Limozin and Sengupta, 2009*; *Biswas et al., 2017*) and images were captured at 19.91 frames per

second using an EMCCD camera (Evolve 512 Delta; Photometrics, Trenton, NJ). Cells that were significantly adhered to the glass bottom were used for imaging. Regions used for analysis were stringently chosen to avoid parts farther than ~100 nm from the coverslip – also called the first branch region (FBR) and the size chosen was 4 × 4 pixels corresponding to ~ $(720 nm)^2$. First, cells were imaged by IRM to take baseline readings following which chemokine (SDF1α) was added. Cells were followed till 30 min with ~2–3 min intervals post-chemokine addition.

For stringent selection of subset of regions, extraction of parameters as well as for visualizing the amplitude of fluctuations, the methodology developed previously (*Biswas et al., 2017*; *Biswas et al., 2019*) was followed. 2048 frames corresponding to ~102 s were captured and used for a single measurement of tension. The analysis involved the calculation of the standard deviation of temporal height fluctuations ($SD_{time}$) and estimating the power spectral density (PSD) by autoregressive technique (probability of covariance or pcov) where the order is chosen to match best with PSD obtained using fast Fourier transform (MATLAB, Mathworks Inc, USA). For deriving mechanical parameters, the PSD is fitted with a Helfrich-based model –

$$PSD\left(f\right) = \frac{4\eta_{eff}Ak_BT}{\pi} \int_{q_{min}}^{q_{max}} \frac{dq}{\left(4\eta_{eff}\left(2\pi f\right)\right)^2 + \left[\kappa q^3 + \sigma q + \frac{\gamma}{q}\right]^2}$$

with four fitting parameters (*Biswas et al., 2017*) using fluctuation-tension (σ), effective viscosity ($\eta_{eff}$), level of confinement (γ), and the effective active temperature (A). The bending rigidity (κ) was fixed at 15 $k_BT$. Membrane fluctuation-tension closely follows membrane frame tension for a large range of tension values (*Shiba et al., 2016*). Membrane fluctuation-tension closely follows membrane frame tension for a large range of tension values (*Shiba et al., 2016*). For brevity, membrane fluctuation-tension has been referred to as 'tension' in the rest of the manuscript. Tension maps were created from PSDs calculated for every pixel which were subsequently fitted and parameters extracted for every pixel. For analyzing the correlation between fluorescence and tension, first, the fluorescence distribution of cells was plotted. Regions of intermediate intensity (around the first peak of the distribution) of a fixed intensity range was filtered and the pixel count of the filtered area was plotted against the average tension in those pixels. Normalization was done by dividing by the maximum value in the time series. Normalized pixel count, thus, reflects how the area with high Piezo1-GFP at the membrane changes while avoiding a small area fraction with very high Piezo1-GFP which might arise from internal structures. To ensure the significance of the observed differences, the statistical test undertaken was the Mann-Whitney U test with Bonferroni correction.

## Piezo1 and actin distribution in live cell imaging

Jurkat cells expressing mCherry-tagged Piezo1 were seeded on ICAM-1-coated dishes. 0.1 µg/ml of recombinant human SDF1α was added to the cells for 15 min prior to imaging. Cell tracking was performed in EVOS FL microscope using the PE-Texas red filter, at an interval of 1 frame per minute, at room temperature. Imaging was performed for a total duration of 15 min.

For Jurkat cells transfected with Piezo1-mcherry and actin-GFP, cells were treated similarly and acquired on Zeiss LSM confocal setup. Z-stack was obtained at 1 µm per stack to capture the entire cell. Time-lapse was imaged at an interval of 30 s per frame for a total duration of 5 min. Confocal imaging was performed at 37 °C.

For particle image velocimetry (PIV) analysis of Piezo1-mCherry-expressing Jurkat cells was performed using the Iterative PIV plugin of FIJI. Time-lapse image stacks were split and stacks of paired subsequent images were formed. Each of these stacks was then corrected for any shift in alignment using the StackReg plugin. Aligned stacks were then subjected to iterative FIJI PIV analysis using the conventional cross-correlation method to measure the flow of fluorescent Piezo1 in chemokine-treated Jurkat cells. PIV post-processing was done with default parameters and PIV magnitude plots were obtained from the plugin.

## Measurement of cell polarity during live-cell imaging

A custom code was written in Fiji (ImageJ) to analyze the cell image time-lapse for the calculation Front/Back (F/B) ratio of actin and Piezo1 expression relative to the direction of cell migration. Briefly, intensity-based thresholding was performed on the actin images to create binary masks of the cells.

Individual cell boundaries were identified in the time-lapse, which were stored in the form of polygon ROIs. The cell ROIs enabled the computation of each cell's trajectory and displacement vectors from the geometrical centroid at every frame of the time-lapse video. The computation of cell polarity involved fitting each cell's ROI to an ellipse and bisecting the ellipse perpendicular to the migration direction. The ratio of total intensities in the leading half and the trailing half of the ellipse was calculated for actin and piezo1 images to get the F/B ratio.

For polarity measurement of fixed confocal images, cell boundaries were similarly identified. Stored ROIs were then split into two halves perpendicular to the long axis of the cell. The polarity index was measured by calculating the difference of total intensities of two halves of the cell divided by the sum of their total intensities. A measurement of 1 shows complete skewing of fluorescent signals to one half of the cell, while a value of 0 shows uniform distribution of fluorescent signals in both cell halves Representative polarity analysis Piezo1 (fixed), Piezo1 mCherry (live), pFAK (fixed), CD11a/LFA1 (fixed), and actin (fixed) of untreated and CCL19-stimulated CD4$^+$ T lymphocytes (*Supplementary file 1*).

For measuring the relative polarity of CD44 and Piezo1, polarity measurement was performed using Fiji as mentioned above. To enable comparison of Piezo1 polarity relative to CD44 polarity, polarity indices [ $\frac{intensity of cell halves (A-B)}{intensities of cell halves (A+B)}$ ] were calculated so as to assign a positive difference, hence polarity, to CD44 (intensity of CD44 in A>B). The same order was used to calculate the polarity index of Piezo1. If CD44 and Piezo1 polarize to the same cell half, then both polar indices will be positive. If, however, Piezo1 polarises to the opposite pole, it will show a negative polarity index with respect to CD44. Paired t-test was used to calculate statistical significance.

## Generation of Piezo1 mCherry and actin-GFP polar plots

For analysis of angular/spatial distribution of Piezo1 mCherry and actin-GFP with respect to direction of cell movement, cell ROIs were applied to the actin and piezo1 channels individually, to find the intensity-weighted center of mass (through all the time-points). Depending on the fluorescence intensity distribution, the intensity-weighted center of mass is shifted away from the geometric centroid and towards the direction of a higher protein distribution. Thus, a vector connecting the centroid (geometric center) to the center of mass gives the direction of polarization of the protein of interest. The relative angle of the polarization vector against the cell displacement vector was calculated for each frame, and a polar histogram of the angles was plotted for actin-GFP and piezo1-mCherry.

## Fixed immunostaining for confocal imaging

Untransfected human CD4$^+$ T lymphocytes were seeded on 12 mm, ICAM-1-coated glass coverslips at a density of about 0.2 million cells per coverslip. Cells were incubated at 37 °C, 5% CO$_2$ for 1.5 hr. 0.5 µg/ml of recombinant human CCL19 was added to each coverslip and incubated for indicated time periods at 37 °C. Post-treatment, cells were fixed in 4% paraformaldehyde for 15 min at room temperature. Blocking was performed in 3% bovine serum albumin in 1 X PBS containing 0.2% Triton X-100 for 50 min at room temperature. Primary rabbit anti-human Piezo1 (Proteintech) was added at a dilution of 1:100 (5 µg/ml) and incubated at room temperature for 1–1.5 hr. Alexa 568-conjugated anti-rabbit IgG was added at a dilution of 1:350 (5.7 µg/ml), and incubated for further 45–50 min. Nuclear staining was performed for 1 min with Hoechst 33342 dye at a concentration of 1 µg/ml in 1 X PBS. All washes between steps were carried out in 1 X PBS. Stained cells were then mounted on Vectashield mounting media and sealed before confocal imaging. For Piezo1 and CD44 co-staining, rabbit anti-Piezo1 and mouse anti-human CD44 were used at a concentration of 5 µg/ml, each. Alexa 488-conjugated anti-rabbit IgG and Alexa 633-conjugated anti-mouse IgG were used at a concentration of 5.7 µg/ml, each.

For actin imaging, LifeAct-TagGFP2 protein was added at a concentration of 0.5 µg/ml, and incubated for 45 min at room temperature before washing. For LFA1 ($\alpha_L\beta2$) and actin co-staining, mouse anti-human CD11a ($\alpha_L$ subunit) was added at a concentration of 5 µg/ml, followed by the addition of anti-mouse Alexa 488-conjugated IgG and Alexa fluor 532 tagged-phalloidin. For LFA1 and Piezo1 co-staining, primary antibodies were added at the above concentrations, with Alexa 568-conjugated anti-rabbit IgG (for Piezo1) and Alexa 488-conjugated anti-mouse IgG (for CD11a/LFA1). Rabbit/mouse anti-phosphorylated FAK (Tyr-397) and mouse Paxillin was used to stain pFAK and paxillin, respectively to detect focal adhesions. These cells were treated with chemokine CCL19 for 10 min prior to fixation.

While doing confocal microscopy, Z-stacks were captured at 0.5 µm intervals. Confocal images are represented are maximum intensity projections of the Z-slices. 3D projection function of Fiji was used to create a 3D rendering of the Z-slices, using the brightest point method of projection.

## TIRF imaging and Ca²⁺-Piezo1 colocalization

$Ca^{2+}$-Piezo1 imaging involved incubating cells with Fluo-3, AM (2.5 µM) in HBSS buffer for 30 min at 37 °C, followed by washing and imaging in RPMI containing 10% FBS, and 25 mM HEPES media. Jurkat cells were transfected with Piezo1 mCherry plasmid and only those expressing Piezo1 mCherry were selected for the study. Dual-color sequential TIRF imaging was employed, with Piezo1 mCherry being imaged at 300 ms (100 frames) and Ca2 + imaging performed using a 10ms exposure time (2000 frames). The short exposure for Ca2 + was used to capture transient local peaks of fluorescence which would otherwise average out at longer exposure times. The maximum projection of the time series was used to detect local peaks using adaptive local thresholding (MATLAB), and a size filter was used to exclude large objects identified with a relatively low $Ca^{2+}$ gradient. For each specific set of settings, the $Ca^{2+}$ signal objects detected were used to calculate the average cluster size and total area of objects. Uniform random numbers were used to simulate clusters with appropriate probability to ensure that the final area of the simulated $Ca^{2+}$ clusters closely matched the real data. The overlap between the real $Ca^{2+}$ signal and Piezo1 objects was compared with the simulated data to test for deviation from a random placement of $Ca^{2+}$ spots. To assess the robustness of the results, different combinations of size filters of Piezo1 and $Ca^{2+}$ objects were used, ranging from 50 to 350 pixels.

## Calpain inhibition

PD150606 (Merck, India), a non-competitive calpain 1 and 2 inhibitors, was used at a concentration of 100 µM. Untransfected $CD4^+$ T lymphocytes were treated with PD150606 for 1 hr at 37 °C prior to treatment of chemokine. Post -chemokine treatment cells were stained either with pFAK as described.

## Focal adhesion kinase inhibition

FAK inhibitor 14 (1,2,4,5-Benzenetetraamine tetrahydrochloride) was used to selectively inhibit FAK activity. Inhibitor was added at a final concentration of 10 µM, 1 hr prior to treatment with recombinant CCL19. Post-treatment with CCL19, $CD4^+$ T lymphocytes were stained with Piezo1 for confocal imaging.

## Piezo1 inhibition

Primary human $CD4^+$ T cells were incubated with 10 µM of GsMTx4 for 1 hr prior to transwell chemotaxis assay in response to recombinant CCL19. Paired t-test was used to calculate statistical significance.

## Statistics

Student t-test was used to calculate significance, unless specified. $p < 0.05$ was considered to be statistically significant.

## Acknowledgements

The study was funded by the Council of Scientific and Industrial Research, India (Grant no. FBR MLP-140). D.G. is a recipient of the Swarnajayanti Fellowship from the Department of Science and Technology, Government of India. The authors express their gratitude for the provision of the Piezo1 constructs by Dr. Charles Cox, Victor Chang Institute of Cardiac Research, Australia.

# Additional information

### Funding

| Funder | Grant reference number | Author |
| --- | --- | --- |
| Council of Scientific and Industrial Research, India | MLP140 | Dipyaman Ganguly |

| Funder | Grant reference number | Author |
| --- | --- | --- |

The funders had no role in study design, data collection and interpretation, or the decision to submit the work for publication.

## Author contributions

Chinky Shiu Chen Liu, Data curation, Software, Formal analysis, Investigation, Visualization, Methodology, Writing - review and editing; Tithi Mandal, Formal analysis, Visualization, Methodology; Parijat Biswas, Methodology; Md Asmaul Hoque, Purbita Bandopadhyay, Bishnu Prasad Sinha, Jafar Sarif, Investigation, Methodology; Ranit D'Rozario, Formal analysis, Investigation, Methodology; Deepak Kumar Sinha, Resources, Software, Formal analysis; Bidisha Sinha, Software, Formal analysis, Supervision, Methodology; Dipyaman Ganguly, Conceptualization, Resources, Formal analysis, Supervision, Funding acquisition, Investigation, Visualization, Writing - original draft, Project administration

## Author ORCIDs

Chinky Shiu Chen Liu ⓘ http://orcid.org/0000-0003-1073-692X
Tithi Mandal ⓘ http://orcid.org/0000-0003-1672-7880
Parijat Biswas ⓘ http://orcid.org/0000-0003-2235-2384
Bidisha Sinha ⓘ http://orcid.org/0000-0001-6449-8205
Dipyaman Ganguly ⓘ http://orcid.org/0000-0002-7786-1795

## Ethics

Human CD4+ T lymphocytes were isolated from peripheral blood of healthy donors through magnetic immunoselection. Peripheral blood cells from healthy human subjects were collected from buffy coats obtained from Tata Medical Center, Kolkata, under a material transfer agreement - the blood donors provided written informed consents. The studies were approved by the Human Ethics Committee of CSIR-Indian Institute of Chemical Biology, India (No. IICB/IRB/2017/03).

Joint Public Review: https://doi.org/10.7554/eLife.91903.3.sa1
Author Response https://doi.org/10.7554/eLife.91903.3.sa2

# Additional files

## Supplementary files

• MDAR checklist

• Source data 1. Source data ile 1 is an Excel file containing all data corresponding to each figure mentioned in the study.

• Source code 1. Source code file 1 includes a zip file containing codes used in Fiji (.ijm files) and MATLAB (.m files) image analysis.

• Supplementary file 1. Representative analyses for calculation of cell polarity of (A) Piezo1 (fixed); (B) Piezo1 mCherry (live); (C) phosphorylated FAK (pFAK, fixed cell); (D) paxillin (fixed cell); (E) CD11a/LFA1 (fixed cell); (F) actin (fixed cell). Region-of-interest identifying cell boundary and polarity indices mentioned. All analyses were performed on Fiji (see methods).

## Data availability

All data included in the study has been uploaded as *Source data 1*. Code used for imaging is provided in *Source code 1*.

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

# Appendix 1

## Appendix 1—key resources table

| Reagent type (species) or resource | Designation | Source or reference | Identifiers | Additional information |
|---|---|---|---|---|
| Transfected construct (*Homo sapiens*) | Piezo1 mCherry; Piezo1-GFP plasmids | From Dr. Charles Cox at Victor Chang Cardiac Research Laboratory, Darlinghurst, Australia (*Cox et al., 2016*) | | |
| Transfected construct (*Homo sapiens*) | pCAG-mGFP-Actin plasmid | Addgene | Plasmid #21948, RRID:Addgene_21948 | |
| Antibody | Piezo1, Rabbit, polyclonal | Proteintech | Cat# 15939–1-AP, RRID:AB_2231460 | 5 µg/ml |
| Antibody | Paxillin, Mouse, Clone 5 H11 | Merck, Sigma Aldrich | Cat# 05–417, RRID:AB_309724 | 5 µg/ml |
| Antibody | phospho-FAK (Tyr 397), Rabbit, polyclonal | Santa Cruz | Cat# sc-11765-R, RRID:AB_653198 | 5 µg/ml |
| Antibody | phospho-Akt (Ser 473), Rabbit, polyclonal | Cell Signalling Technolgies | Cat# 9271, RRID:AB_329825 | 5 µg/ml |
| Antibody | Akt1, Goat, polyclonal | Santa Cruz | Cat# sc-1618, RRID:AB_630849 | 5 µg/ml |
| Antibody | anti-human APC CD44, Mouse, Clone G44-26 | BD Bioscience | Cat# 560890, RRID:AB_398683 | 5 µg/ml |
| Antibody | anti-human FITC CD11a/LFA-1, Mouse, Clone G43-25B | BD Pharmingen | Cat# 555379, RRID:AB_395780 | 5 µg/ml |
| Antibody | anti-mouse IgG Alexa Fluor488, Goat | Thermo Fisher, Invitrogen | Cat# A-11029, RRID:AB_2534088 | 1: 350 dilution |
| Antibody | anti-rabbit IgG Alexa Fluor 568, Goat | Thermo Fisher, Invitrogen | Cat# A-11036, RRID:AB_10563566 | 1: 350 dilution |
| Antibody | anti-goat IgG Alexa Fluor 647, Chicken | Thermo Fisher, Invitrogen | Cat# A-21469, RRID:AB_2535872 | 1: 350 dilution |
| Antibody | Human CD4 microbeads, T cells | Miltenyi Biotec | Cat# 130-097-048 | 1: 350 dilution |
| Peptide, recombinant protein | LifeAct Tag-GFP2 protein | Ibidi | Cat# 60112 | 0.5 µg/ml |
| Peptide, recombinant protein | Recombinant Human MIP-3β (CCL19) | Peprotech | Cat# 300-29B | 0.5 µg/ml |
| Peptide, recombinant protein | Recombinant Human ICAM-1 | Peprotech | Cat# 150–05 | 4 µg/ml |
| Peptide, recombinant protein | Recombinant Human SDF-1α (CXCL12) | Peprotech | Cat# 300–28 A | 0.1 µg/ml |
| Peptide, recombinant protein | Recombinant human fibronectin (RetroNectin) | Takara | Cat# T202 | 10 µg/ml |
| Chemical compound, drug | Calpain Inhibitor, PD 150606 | Merck, Sigma Aldrich | Cat# D5946 | 100 µM |
| Chemical compound, drug | FAK Inhibitor 14 | Merck, Sigma Aldrich | Cat# SML0837 | 10 µM |
| Peptide, recombinant protein | Piezo1 Inhibitor, GsMTx4 | Tocris | Cat# 4912 | 10 µM |
| Commercial assay or kit | Lonza P3 Primary Cell 4D-Nucleofector X Kit L | Lonza | Cat# V4XP-3024 | |
| Sequence-based reagent | Human Piezo1 Forward primer | IDT | | 5′-CCC AAG TGG AGC TCA GGC CC-3′ |

*Appendix 1 Continued on next page*

*Appendix 1 Continued*

| Reagent type (species) or resource | Designation | Source or reference | Identifiers | Additional information |
|---|---|---|---|---|
| Sequence-based reagent | Human Piezo1 Reverse primer | IDT | | 5'-GGG CCA GGG ACA GGC AGA AG-3' |
| Sequence-based reagent | Human Piezo2 Forward primer | IDT | | 5'-CAT CTA CAG ACT GGC CCA CCC G-3' |
| Sequence-based reagent | Human Piezo2 Reverse primer | IDT | | 5'-AGA GCA CAG TGA GGC GGT CA-3' |
| Sequence-based reagent | Human 18 S primer | IDT | | 5'-GTA ACC CGT TGA ACC CCA TT-3' |
| Sequence-based reagent | Human 18 S primer | IDT | | 5'-CCA TCC AAT CGG TAG TAG CG-3' |
| Sequence-based reagent | EGFP control mission esiRNA | Merck, Sigma Aldrich | Cat# EHUEGFP | |
| Sequence-based reagent | Human Piezo1 mission esiRNA | Merck, Sigma Aldrich | Cat# EHU135531 | |
| Cell line (*Homo sapiens*) | Jurkat cell line | From Dr. Santusabuj Das at National Institute of Cholera and Enteric Diseases (NICED), Kolkata, India. ATCC | TIB-152 | |
| Other | Fluo-3, AM (Calcium indicator) | Thermo Fisher, Invitrogen | Cat# F1241 | |
| Other | Hoechst 33342 | Merck, Sigma Aldrich | Cat# 14533 | |
| Other | Cell Trace Violet | Thermo Fisher, Invitrogen | Cat# C34557 | |
| Other | Cell Trace CFSE | Thermo Fisher, Invitrogen | Cat# C34554 | |
| Other | Alexa Fluor 532 Phalloidin | Thermo Fisher, Invitrogen | Cat# A22282 | |
| Software, algorithm | Fiji | Open Source | | https://fiji.sc/ |
| Software, algorithm | MATLAB 2020b | IISER, Kolkata academic license | | |

