## [Editor Report · eLife assessment]

This study provides **useful** insights into the subcellular localization, interaction with integrins, and functional importance of the cell surface receptor Piezo1 in migrating human T-cells. Whether Piezo1 is critically sensing mechano-physical cues during T-cell migration is however not well supported by direct experimental evidence. The data collected is **solid** otherwise.

---

## [Referee Report · Joint Public Review]

This work by Liu CSC et al. is an extension of the author's previous work on the role of Piezo1 mechano-sensor in human T cell activation. In this study, the authors address whether Piezo1 plays a role in T-cell chemotactic migration.

The authors used CD4+ T cells or Jurkat T cells to test the effects of siRNA-mediated depletion of Piezo1 on chemotactic migration. They establish that Piezo1 is implicated in chemotactic migration, although the effects of depletion are relatively moderate.

They show that Piezo1 is redistributed to the leading edge of T-cells.

They identify that relocation of Piezo1 to the leading edge follows an increase in membrane tension.

In Piezo-1 depleted cells, they observe a moderate reduction of LFA-1 polarity. With the use of specific inhibitors, they propose Piezo1 activation to be downstream of focal adhesion formation and upstream of calpain-mediated LFA-1, integrin alpha L beta 2, or CD11a/CD18 recruitment at the leading edge.

Strengths:

Together with their 2018 paper, this study presents Pieszo1 as a regulator of T-cell activation, implicating it as a player in the coordination of the chemotactic immune response.

Weaknesses:

Most of the effects observed are relatively modest. The authors did not challenge the cells with various physico-mechanical conditions to see when Piezo-1 might become really important. For instance, there are no experiments that expose T cells to varying counter-acting forces to see how piezo1 might affect migration.

Technical weaknesses:

The authors state that "these high tension edges are usually further emphasized at later time points", but after ten minutes the median tension and tension (Figure 2C and Supplementary Figure 2C respectively) reduce down to the pretreatment time point. It would be clearer if the author stated within which timeframe the tension edges are "further emphasized".

Figures 3 and 4 - The author states the number of cells quantified from the images, but it is not clear whether the data is actually from 3 biological replicates.

Some of the data has no representative images or videos included. There is no video in the supplementary for Figures 1 A and B. There are no representative images of transwell migration assay in Figures 1 D and E.

---

## [Author Response]

The following is the authors’ response to the original reviews.

Response to comments of editor/s:

• With regard to the comments on nonavailability of representative images/videos for Figures 1 A and B, in the revised manuscript we have added a representative video of GFP (-) and GFP (+) tracks in Animation 1.

Response to comments of reviewer 2:

• With respect to the concern on figure 1, we have changed ‘% CD4+ T cell Migration’ to ‘% Proportion CD4+ T cell migration’ in Figures 1D & 1E in the revised manuscript. We also labelled the upper and lower panels of Figure 1I as ‘Untreated’ and ‘SDF1α’ respectively.

Response to comments of reviewer 1:

• With regard to the concern that ‘The transfection alone with siRNA may cause the lack of polarity’, we have added comparison of 2D migration MSD between control EGFP siRNA and Piezo1 siRNA-transfected CD4+ T cells as Figure 1- figure supplement 1E.

• We have added new references as Chen et al. (Neuron, 2018) and Yao et al. (Sci Adv, 2022), with respect to PIEZO1 association with focal adhesions.

• With regard to the concerns around co-localization of Piezo1 and focal adhesions, we have added a representative image of Piezo1 and pFAK co-localization upon treatment of chemokine in revised Figure 3 - figure supplement 1C. We have also used an additional focal adhesion marker, paxillin, to show that focal adhesion formation is not affected by Piezo1 KD (Revised Fig. 3E-3H). Upon comparing the mean pFAK and paxillin intensities, we observed no difference in Control and Piezo1 knockdown CD4+ T cells (Figure 3 - figure supplement 1A, B).

• All the minor concerns and suggestions have been taken care of in the revised manuscript.Source code 1